# Energy-Efficient Random Variate Generation via Compressed Lookup Tables

**Johann Ukrow**[*], **Anna Kazachkova**[*], **Nicolas Alder, Sven Köhler,**
**Rainer Schlosser & Ralf Herbrich.**
Hasso Plattner Institute
University of Potsdam
Potsdam, Germany
`{johann.ukrow,anna.kazachkova,nicolas.alder,sven.koehler,`
`rainer.schlosser,ralf.herbrich}@hpi.de`

## Abstract

Generating (pseudo-)random variates lies at the core of probabilistic machine learning and prediction algorithms and yet remains a major bottleneck due to its high computational and energy cost. In this paper, we introduce a general and scalable sampling strategy that enables fast and energy-efficient random variate generation from arbitrary distributions. Our approach is based on compressed lookup tables (cLUT) combined with a fast index sampling scheme. Using only a handful of fast and energy-efficient compute operations on simple array structures, we achieve superior speed, energy efficiency, and precision at near-optimal entropy cost compared to state-of-the-art techniques. Microbenchmarking our approach with a C implementation shows up to 40% savings in time and 50% in energy compared to state-of-the-art approaches. Compared to commonly employed Python samplers, we achieve a $100\times$ time improvement.

## 1 Introduction

Sampling from probability distributions is a fundamental yet computationally expensive operation in machine learning. In representation learning and in broader machine learning, sampling underpins core methods such as variational autoencoders (Kingma and Welling, 2022), contrastive learning with negative sampling (Chen et al., 2020), diffusion-based generative models (Ho et al., 2020b), and probabilistic inference techniques such as Bayesian deep learning (Sommer et al., 2025). While the quality and efficiency of sampled variables directly shape the expressiveness and scalability of learned representations, sampling costs often remain a primary barrier to scalability and widespread deployment. In this paper, we address this bottleneck by introducing a novel, efficient sampling approach for arbitrary distributions. Our method achieves 10-100$\times$ speedups and up to 60% reduction in energy consumption compared to commonly employed approaches, significantly reducing the resource-intensity of many machine learning tasks.

The continued deployment of machine learning methods in data centers, cloud devices, and user appliances alike is accompanied by increased concerns about the growing energy demand of the field (International Energy Agency, 2025; Gadepally, 2025). Countermeasures include reducing the carbon-intensity of the electricity supply or shifting training and inference to times or physical locations with a higher share of renewable energy sources throughout the day (Yang et al., 2023; Wiesner et al., 2023). However, we argue that reducing the energy demand of the operations themselves is worthwhile, with emphasis on frequently performed actions like sampling. Motivating us to perform extensive energy measurements in addition to speed measurements, comparing our method to classical, widely used approaches and recent state-of-the-art advances.

On digital computers, sampling from arbitrary probability distributions is reduced to sampling from finite discrete distributions due to fundamental constraints of finite precision and memory. All probability distributions, whether continuous or infinite discrete, must be discretized for computational

---

[*]Equal contribution.

implementation (see Appendix B for discretization techniques). Standard sampling algorithms in widely used libraries (NumPy, PyTorch, JAX) assume infinite precision arithmetic, where computation can be performed with arbitrarily precise real numbers (Shamos 1978; Devroye 1986, Chapter 2, p.1, Assumption 1). Additionally, they assume the ability to generate infinitely precise samples from the real unit interval (Devroye 1986, Chapter 2, p.1, Assumption 2). However, actual implementations rely on finite floating-point representations and don't have access to exact samples from the real unit interval, causing generated distributions to deviate from their intended target distributions in uncontrolled ways. These deviations are often intractable to quantify, precluding theoretical guarantees about sampling accuracy. To address this issue, our proposed sampling method features controllable precision and exactly represents target distributions with clear theoretical guarantees.

**Problem formulation**   In what follows, we will describe a novel method to generate random variates from any finite discrete distribution, represented by $n$ probabilities $p_1, \ldots, p_n \in [0, 1]$ of the corresponding $n$ outcomes $x_1, \ldots, x_n \in \mathcal{X}$. As we denote no constraints on the structure of $\mathcal{X}$ the outcomes can be of arbitrary types, such as real numbers, strings, pointers to more complex data structures, or any mixture of those. Our objective is to make this generation process fast, energy efficient, and adjustable to an arbitrary and controllable precision.

**Our contributions**   This work introduces a new sampling approach for arbitrary distributions based on operations with lookup tables. Besides being a generic method for efficient and arbitrary precise sampling, our approach is especially suitable for situations where floating-point operations are either unavailable or too error prone, and situations with a low power supply. We summarize our contributions as follows:

1. We propose a novel a random variate generator based on compressed lookup tables (cLUT), optimized for highly efficient sampling. We introduce a lossless compression strategy for compact representations of distributions achieving an exponential compression ratio.

2. We compare cLUT against state-of-the-art approaches in terms of speed, energy efficiency, memory usage, and entropy efficiency. It runs 30-40% faster and saves 25-50% energy in a diverse set of distributions. For larger distribution sizes, it performs particularly well.

3. We benchmark cLUT against standard sampling routines from widely used Python machine learning libraries. cLUT achieves up to 10-100× acceleration in speed. Furthermore, we illustrate the impact of our approach in real-world machine learning applications by showcasing that cLUT substantially reduces the execution time and energy consumption of the exemplary TrueSkill application.

## 2   RELATED WORK

Sampling methods are classically divided into two categories: *exact methods*, which produce samples from the target distribution $\mathbf{p}$ as specified, and *approximate methods*, which generate samples from a distribution $\tilde{\mathbf{p}}$ that only approximately matches the desired distribution, i.e., $\tilde{\mathbf{p}} \approx \mathbf{p}$. Note that our approach is exact.

**Exact methods**   Knuth and Yao (1976) established the theoretical foundation for exact discrete sampling using discrete distribution generating trees. Their seminal result shows that any optimal sampling algorithm requires between $H(\mathbf{p})$ and $H(\mathbf{p}) + 2$ bits per sample, where $H(\mathbf{p}) = \sum_i -p_i \log_2(p_i)$ is the Shannon entropy. While entropy-optimal, discrete distribution generating trees typically require exponential memory in the distribution precision. Lumbroso (2013) overcame this limitation for uniform and Bernoulli distributions with a linear-memory implementation, but the approach does not generalize to arbitrary distributions. The generic interval algorithm (Hao and Hoshi, 2006) achieves linear memory usage while consuming at most $H(\mathbf{p}) + 3$ bits per sample. However, implementations require expensive binary searches at each sampling step, limiting practical efficiency (Devroye and Gravel, 2020; Uyematsu and Li, 2003). Saad et al. (2020) presented the FLDR algorithm that combines entropy-optimal sampling with rejection sampling, achieving an upper bound of $H(\mathbf{p}) + 6$ bits per sample. **?** improved this to $H(\mathbf{p}) + 2$ bits with faster sampling speed for the ALDR algorithm, though at a higher memory cost. Building on Marsaglia (1963), Marsaglia et al. (2004) proposed compressed lookup tables for discrete sampling. However, their compression scheme requires conditional branching and searches across multiple tables during sampling,

reducing efficiency. In contrast, our approach uses a single compressed table with direct indexing, eliminating conditional overhead.

**Approximate methods**   Most samplers for discrete and continuous distributions used in practice are so-called approximate samplers (for an introduction, see Schwarz, 2011). These methods typically rely on the assumptions of the *real Random Access Machine* (RAM) model (Shamos, 1978, computations can be performed with arbitrarily precise real numbers), and the assumption of having infinitely precise uniform random generators (Devroye, 1986), which cannot be fully realized on digital computers. For a comprehensive overview, Devroye (1986) presents the mathematical foundations of random sampling and details numerous approximate samplers built on the real RAM model. As noted by Draper and Saad (2025), implementations consequently suffer from multiple sources of approximation error and are often inefficient in their use of bits, since generating a single uniform random variable typically already consumes 32 or 64 bits. A widely employed general approximate sampler is the Alias method (Walker, 1974), which preprocesses distributions into probability and alias arrays, enabling sampling via one uniform random variable and a single coin flip (see Schwarz (2011) for a detailed explanation). While being fast, it produces approximate samples and lacks controllable error bounds. Similarly, the Index method (Chen and Asau, 1974) uses preprocessed index tables to guide inversion-based approximate sampling, but still requires expensive search operations.

In contrast, our proposed method does not rely on the real RAM assumption or on access to arbitrarly precise random samples from the unit interval. It achieves exact sampling while remaining highly entropy-efficient, using close to the minimum number of bits required to represent the target distribution (see Figure 4).

## 3   APPROACH

Our approach is based on the idea of lookup tables, reusing precomputed results, while conserving memory requirements and memory accesses as detailed in this section. A schematic of the sampling pipeline is given in Figure 1a.

**Naive approach**   We will describe a method to generate random variates from any finite discrete distribution, represented by $n$ probabilities $p_1, \ldots, p_n \in [0, 1]$ of the $n$ outcomes $x_1, \ldots, x_n \in \mathcal{X}$. Ideally, we would construct a lookup table containing duplicates of each outcome proportional to its probability:

$$\frac{\text{occurrences of } x_i \text{ in table}}{\text{table size}} = p_i. \tag{1}$$

Sampling would then reduce to uniformly selecting a random table index $I \sim \text{Uniform}\{1, \ldots, N\}$ and returning $S = \text{Table}[I]$, where $N$ is the table size. See Figure 1b for an examplary 'naive' lookup table.

**Memory constraints**   In practice, memory constraints bound the table size $N$, limiting representable probabilities to multiples of $1/N$. Approximating probabilities to precision $b$ bits requires quantizing each $p_i$ to $f_i = \text{round}(p_i \cdot 2^b)$, yielding frequencies $\mathbf{f}=(f_1, \ldots, f_n)$ and a table of size $N = \sum_i f_i = 2^b$ (see the Appendix C for rounding schemes and error analysis). While approximation error decreases logarithmically with $b$, memory requirements grow exponentially, making high-precision sampling prohibitive. Our approach also handles continuous and infinite discrete distributions through discretization techniques detailed in the Appendix, Section B. The basis of our main approach is a lossless compression strategy for the lookup tables and the following sampling scheme.

**Compression scheme**   To tackle the prohibitive memory requirements of lookup tables, we propose to use the following compression scheme. Intuitively, the compressed lookup table can be viewed as a two-dimensional array consisting of $r + 1$ rows and $2^c$ columns, with $r, c \in \{0, \ldots, b\}$ satisfying $2^{r+c} = 2^b = N$. Each row $i$ of the first $r$ rows corresponds to a frequency of $2^{r-i}$, where row indices run from 1 to $r$. The $r+1$-th row corresponds to the same frequency as the $r$-th row, namely $2^{r-r} = 1$. For an exemplary compression, see Figure 1b. This lossless compression scheme

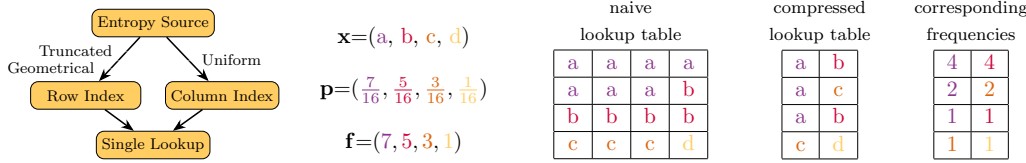

(a) Schematic of sampling.         (b) Example (compressed) lookup table.

Figure 1: **a.** Schematic of generating a single sample using our approach: i.i.d. $\mathrm{Ber}(0.5)$ bits are drawn from an entropy source to compute a row and a column index, yielding in a single lookup on the precomputed table. **b.** Illustration of the precomputation step: A naive and compressed lookup table for an example distribution given by $\mathbf{x}=(a, b, c, d)$ and $\mathbf{p}=(\frac{7}{16}, \frac{5}{16}, \frac{3}{16}, \frac{1}{16})$. The naive lookup table (left table) contains each value according to its frequency $\mathbf{f}=(7, 5, 3, 1)$ at a precision of $b = 4$ bits. The compressed lookup table (middle table) stores the same distribution when considering the geometric frequency scheme (right table). For example, the frequency of "a" is given by the compressed lookup table as $4 + 2 + 1 = 7$ and thus equals the frequency of "a" in the naive lookup table.

preserves the total frequencies

$$\sum_{i=1}^{r} 2^{i-1} \cdot 2^c + 2^c = (2^r - 1 + 1) \cdot 2^c = 2^{r+c} = N,$$

while drastically decreasing the size of the lookup table. Compressing a naive lookup table with $N = 2^{r+c}$ entries to a compressed lookup table with $(r + 1) \cdot 2^c$ entries (organized in $r + 1$ rows and $2^c$ columns) yields a compression ratio of $\rho = 2^r/(r + 1)$.

In the example of Figure 1b, the compression ratio would be $\rho = 2^3/(3 + 1) = 2$, which yields a compressed table half the size of the naive table. The compression ratio $\rho$ improves exponentially with $r$, up to a linear correction factor of $r + 1$, with the concrete values of $r$, and therefore of $\rho$, depending on the frequencies $\mathbf{f}$. Intuitively, better compression corresponds to an increase in the number of rows (larger $r$) accompanied by an exponential decrease in the number of columns, resulting in a "taller" and much "narrower" lookup table. This compression scheme is always possible for lookup tables of size $2^b$. To see that, note that the compressed and the naive lookup table coincide for the choice of $r=0$ and $c=b$.

**Sampling step** To generate a sample $S \in \mathcal{X}$ using a compressed lookup table, we generate two indices independently: a row index $I \in \{1, \ldots, r+1\}$ and a column index $J \in \{1, \ldots, 2^c\}$. We sample the index $I$ according to a truncated geometric distribution, and the column index $J$ uniformly:

$$\mathbb{P}(I = i) = \max(2^{-i}, 2^{-r}) \text{ for } i = 1, \ldots, r+1, \quad \text{and} \quad \mathbb{P}(J = j) = 2^{-c} \text{ for } j = 1, \ldots, 2^c.$$

Therefore, we sample a table-index $(I, J) = (i, j)$ with probability $2^{-\min(i,r)-c}$. The column index $J$ can be efficiently sampled using any uniform sampler. The row index $I$ can also be sampled extremely efficient using the entropy optimal procedure detailed in Algorithm 1 in lines 2-8. A sample is then generated by returning the value stored in the compressed lookup table at that index:

$$S = \texttt{compressedTable}[I, J].$$

**Preprocessing step** Before sampling, we must construct the compressed table. Conveniently, we do not have to construct the uncompressed lookup table, which could induce severe memory issues. We rather construct the compressed lookup table directly from the binary expansion of the frequencies $f_i$. A value $x_i$ appears in row $j$ if and only if the $j$-th bit $f_i^{(j)}$ of $f_i$ is one. The frequencies $\mathbf{f}$ can be adjusted to sum to exactly $2^b$ by using a sum-preserving rounding scheme, making our sampling procedure rejection-free. Although the total probability mass and relative ratios are preserved in the compressed lookup table, the number of active bits across the binary representations of the $f_i$ may differ, which results in rows of unequal width in the initial construction of the compressed lookup table. To improve the sampling speed, we ensure that all rows have uniform width as detailed in Algorithm 2 and Figure 2.

---

**Algorithm 1** Sampling from compressed lookup tables

---

**Require:** number of samples $K$, compressed lookup table `compressedTable` of size $(r+1)\times 2^c$
**Ensure:** array of samples $S$
  1: **for** $k = 1$ **to** $K$ **do**
  2:     $I \leftarrow 1$
  3:     **while** $I < r + 1$ **do**   *// Sample row index geometrically:*
  4:         **if** `randomBit()` $= 1$ **then**
  5:             **break**
  6:         **end if**
  7:         $I \leftarrow I + 1$
  8:     **end while**
  9:     $J \leftarrow \mathrm{Uniform}\{1, \ldots, 2^c\}$   *// Sample column index uniformly.*
 10:     $S[k] \leftarrow$ `compressedTable`$[I, J]$   *// Generate a sample from the distribution.*
 11: **end for**
 12: **return** $S$

---

| | distribution | binary expansion | | | initial compressed lookup table | | | | | | rectified compressed lookup table | | | |

distribution

$\mathbf{x} = (a, b, c, d, e)$

$\mathbf{p} = (\frac{14}{32}, \frac{6}{32}, \frac{7}{32}, \frac{3}{32}, \frac{2}{32})$

$\mathbf{f} = (14, 6, 7, 3, 2)$

binary expansion

$(f_1)_2 = 1110$
$(f_2)_2 = 0110$
$(f_3)_2 = 0111$
$(f_4)_2 = 0011$
$(f_5)_2 = 0010$

initial compressed lookup table

| Frq. | | | | | |
|---|---|---|---|---|---|
| 8 | a | | | | |
| 4 | a | b | c | | |
| 2 | b | c | d | e | a |
| 1 | c | d | | | |
| 1 | | | | | |

rectified compressed lookup table

| Frq. | | | | |
|---|---|---|---|---|
| 4 | a | b | c | a |
| 2 | b | c | d | e |
| 1 | c | d | a | a |
| 1 | a | a | a | a |

Figure 2: The initial and final compressed lookup table for an example distribution given by $\mathbf{x}$=(a, b, c, d, e) and $\mathbf{p}$=$(\frac{14}{32}, \frac{6}{32}, \frac{7}{32}, \frac{3}{32}, \frac{2}{32})$. In a first step, the table is filled according to the binary expansion of the frequencies (left table). Then, the table is rectified by moving entries from higher to lower rows while doubling (right table). For example, the "a" in the top row corresponding to a frequency of 8 (blue), is replaced by one "a" in the second row, which corresponds to a frequency of 4, and 4 "a"'s in the bottom rows, which correspond to a frequency of 1, while the "a" in the third row ( frequency of 2) is replaced by further two "a" in the bottom rows. The total frequency of each value is preserved, and the rows have equal length. In this case, $b = 5, r = 3$, and $c = 2$.

## 4 EVALUATION

To demonstrate the advantage of our sampling method, we compared it to state-of-the-art sampling methods in five experiments.

**Evaluation Setup**   All measurements were taken on a standard laptop equipped with an Intel i7-1255U CPU and 16 GiB DDR4 memory running Ubuntu Linux.

Modern CPUs provide hardware counters that monitor the current power and energy demand. On the x86_64 platform, *Running Average and Power Limit* (RAPL; David et al. 2010) counters provide energy readings at a 1 ms resolution. RAPL is organized into different power domains, representing different parts of the system. For this work, we focus on the CPU domains *cores* and *package* (pkg). The latter includes the former and additionally other parts of the CPU socket, such as caches and the memory controller. There are several factors that make energy measurements noisy, apart from default hardware noise. They include background activities, battery charging, artificial noise against side-channel attacks for security reasons (Lipp et al., 2021), etc. We limit these influences by disabling CPU security features, keeping the laptop charged, providing additional warm-up rounds, and measuring multiple iterations. Additionally, we set a constant CPU frequency and CPU core to get meaningful energy readings, as detailed in the Appendix, Section J. We evaluate all methods (both in Python and in C) on a fixed set of synthetically generated distributions of sizes $n \in [10^1, 10^7]$ drawn from exponential distributions with varying parameters to span a broad range of entropy values, with zero probabilities removed. The bit precision is set to $b = 16$ for $n \in [10^1, 10^4)$, $b = 20$ for $n \in [10^4, 10^6)$ and $b = 23$ for $n \in [10^6, 10^7]$. Evaluations on further distributions are in the Appendix, Section G.

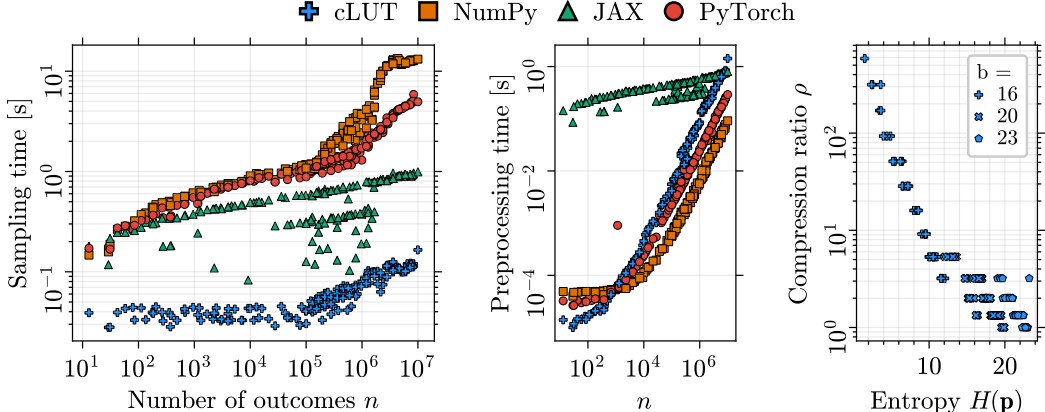

Figure 3: Comparison of our cLUT approach with standard sampling methods from popular machine learning libraries (NumPy, JAX, PyTorch). Shown are (1) the average wall time (in seconds) to generate $10^7$ samples from distributions of varying sizes $n \in [10^1, 10^7]$, (2) the preprocessing time, and (3) the compression ratios $\rho$ of the cLUT algorithm. Distributions are discretized Exponential distributions with varying parameters to cover a broad range of entropies, with zero probabilites excluded. The plots are shown on a log-log scale. Each measurement was repeated ten times and averaged.

Table 1: Average wall time (in seconds, mean $\pm$ std) for generating $10^7$ samples and the preprocessing step in Python. Evaluated in two subsets of the distributions from Figure 3, split by size.

| # Outcomes: | $n \in [10^1, 10^5)$ | | $n \in [10^6, 10^7)$ | |
|---|---|---|---|---|
| Method | Sampling time (s) | Preprocessing time (s) | Sampling time (s) | Preprocessing time (s) |
| NumPy | $0.6680 \pm 0.2650$ | $0.0001 \pm 0.0001$ | $9.6248 \pm 3.5823$ | $0.0308 \pm 0.0202$ |
| PyTorch | $0.6073 \pm 0.2436$ | $0.0003 \pm 0.0006$ | $3.3768 \pm 1.0615$ | $0.1028 \pm 0.0655$ |
| JAX | $0.3647 \pm 0.1277$ | $0.2898 \pm 0.0894$ | $0.7982 \pm 0.1815$ | $0.6528 \pm 0.0948$ |
| cLUT | $0.0374 \pm 0.0051$ | $0.0006 \pm 0.0011$ | $0.1016 \pm 0.0129$ | $0.3925 \pm 0.2219$ |

**Sampling speed in Python**    We benchmarked our method against standard discrete sampling routines from widely used Python machine learning libraries: `RandomGenerator.choice()` from NumPy, `multinomial()` from PyTorch, and `random.choice()` from JAX (Harris et al., 2020; Paszke et al., 2019; Bradbury et al., 2018). As shown in Figure 3, our method achieves a 10–100× speedup across a wide range of distributions. The performance advantage is most pronounced for distributions with a large number of outcomes: Table 1 reports a 10× improvement for distributions with $10^4$ to $10^5$ entries, with the speedup growing to over 100× already for distributions with $10^6$ to $10^7$ entries. This is particularly relevant when targeting high-precision and high-diversity random variate generation: a 16-bit data type can already represent 65,536 distinct values, whereas 32-bit and 64-bit types can represent vastly more (over $10^9$ and $10^{19}$, respectively), making conventional sampling increasingly inefficient. Furthermore, one should note that the samplers in NumPy, PyTorch, and JAX build on the Inversion method (Devroye, 2006) and produce distributions that are only approximately similar to the desired distribution, whereas our proposed method produces exactly the specified distribution.

For evaluations including graphics processing units see the Appendix, Section F, and Figure 7.

**Sampling speed compared to SOTA implementations in C**    We compare our cLUT sampler to the following state-of-the-art sampling methods: the Alias method (Walker, 1974), ALDR, and FLDR (Draper and Saad, 2025). All samplers are implemented in C, as it provides a lower execution overhead compared to, e.g., Python. This allows for proper assessment of the actual costs of the algorithm. To ensure comparability, we apply the same degree of fair but not over-engineered opti-

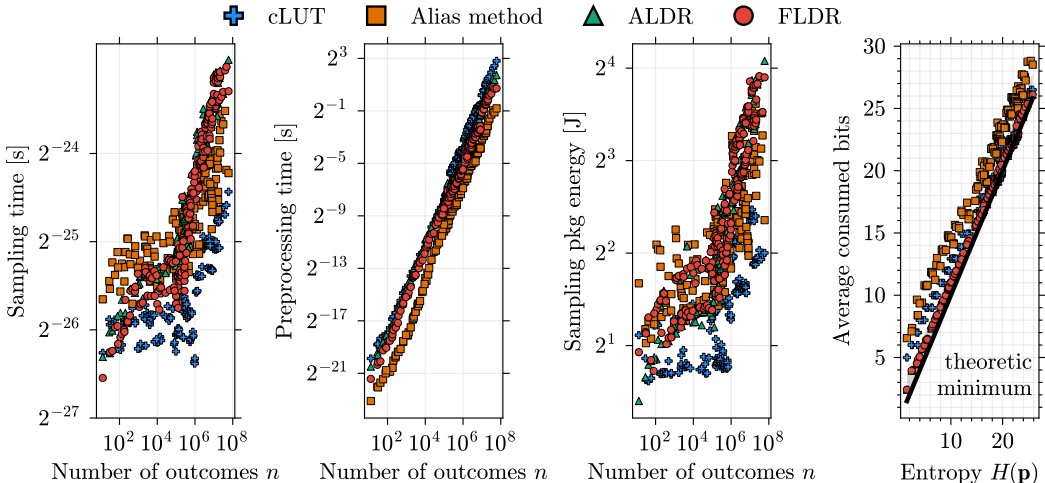

Figure 4: Comparison of our cLUT approach with existing state-of-the-art sampling methods in C. Shown are (1) the wall time required for generating a single sample (averaged over $10^7$ repetitions) and (2) preprocessing (averaged over 10 repetitions), as well as (3) the cumulative energy demand of the CPU socket for generating $10^7$ samples. Time and energy are shown on a log-log scale. The fourth subfigure shows the average consumed bits per sample from the entropy source.

Table 2: Average energy demand, wall time, and power draw of a single sampling operation in C. The power draw series is computed by dividing the energy series by wall time. It averages over all CPU instructions of a sampling iteration. High variance in entropy and distribution size results in the high standard deviation here. Shown are distributions from Figure 4, split by size.

| # Outcomes: | $n \in [10^3, 10^4)$ | | | $n \in [10^7, 10^8)$ | | |
|---|---|---|---|---|---|---|
| **Method** | **Energy (nJ)** | **Time (ns)** | **Power (W)** | **Energy (nJ)** | **Time (ns)** | **Power (W)** |
| ALDR | $263.451 \pm 50.309$ | $22.431 \pm 1.250$ | $11.836 \pm 2.694$ | $1223.520 \pm 194.457$ | $102.792 \pm 12.066$ | $12.168 \pm 2.899$ |
| Alias method | $319.804 \pm 72.045$ | $26.803 \pm 2.934$ | $12.156 \pm 3.549$ | $887.653 \pm 185.627$ | $55.946 \pm 13.709$ | $16.502 \pm 3.897$ |
| FLDR | $290.223 \pm 47.274$ | $21.268 \pm 2.373$ | $14.031 \pm 3.770$ | $1214.382 \pm 177.125$ | $101.404 \pm 9.702$ | $12.195 \pm 2.753$ |
| cLUT | $199.233 \pm 38.579$ | $15.475 \pm 1.689$ | $13.271 \pm 4.091$ | $450.155 \pm 74.604$ | $33.026 \pm 4.880$ | $14.188 \pm 4.188$ |

mization across these methods. We avoid multi-threading or (auto) vectorized code and use identical compiler flags. All methods use the identical entropy source.

For our experiment, we distinguish between the preprocessing phase (ten repetitions) and the actual sampling operation (ten million repetitions). The latter can be performed quickly and repeatedly after the higher, one-time upfront cost. Figure 4 shows our results, indicating that our cLUT method samples consistently faster for all distributions than our competitors in terms of sampling time. Table 2 shows mean and standard deviation of the sampling time on two representative subsets of the distributions ($n \in [10^3, 10^4)$ and $n \in [10^7, 10^8)$).

**Energy consumption compared to SOTA implementations in C**   To demonstrate the energy-saving potential of our approach, we compare the energy demand of all implementations. Due to space restrictions, Figure 4 only shows the energy demand of the sampling operation across the entire RAPL package domain (CPU socket and memory controller), which is representative of the other measurements. Again, our cLUT approach works best across all sizes.

Although the energy demand roughly follows the same trend as the required time, the scale is not linear. This is because time is not an accurate indicator of the energy required by complex real-life computing systems. Rather, energy is the integral of the dynamic power demand over time.

Our cLUT's single index-based memory lookup requires fewer switching transistors in the memory subsystem, compared to, e.g., ALDR with multiple memory accesses to a flattened search tree.

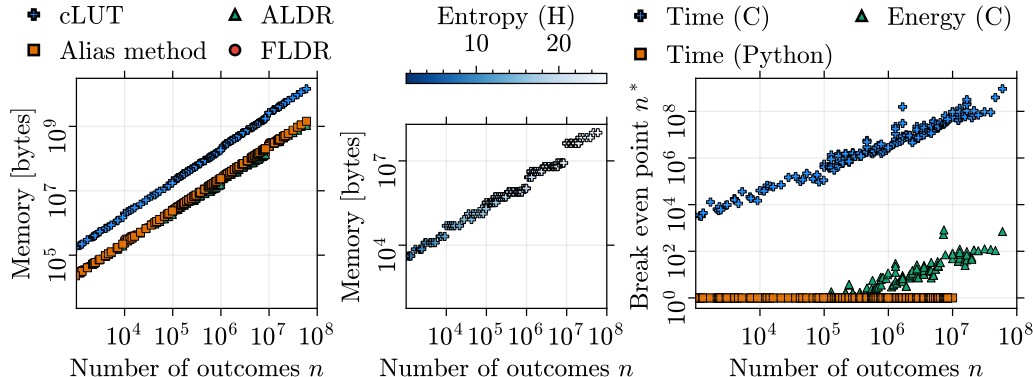

Figure 5: Comparison of cLUT with state-of-the-art sampling methods in terms of memory usage and break-even analysis: (1) peak memory usage for all methods (including preprocessing), (2) memory usage of compressed cLUT table, and (3) break-even analysis against the Alias method. The break-even point $n^*$ is the minimum number of samples needed for cLUT to offset its preprocessing overhead relative to the Alias method (in terms of sampling time or energy consumption).

**Memory usage and preprocessing overhead** We measure peak memory usage for all approaches and perform a break-even analysis for sampling time and energy consumption compared to the commonly used Alias method. *Peak memory usage* refers to the maximum amount of memory utilized by a program during execution. As shown in Figure 5, the cLUT approach consumes slightly more memory at peak times than other state-of-the-art algorithms. However, the constructed compressed lookup tables and therefore the memory usage after preprocessing is relatively small, especially for low-entropy distributions due to high compression ratios (see Figure 5, middle Figure; and compare Figure 3 for compression ratios). A break-even analysis against the Alias method shows that this overhead is offset after a reasonable number of sampling iterations.

As shown in Figure 4, the preprocessing phase (look-up table creation) scales log-linear with the distribution size across all investigated methods. Our cLUT method shows the highest time demand for the preprocessing phase and the Alias method the lowest one. Thus, our approach requires more sampling operations to offset its higher initial costs, but is then more time efficient, especially for larger distributions. As shown in Figure 5, the break-even point $n^*$ for sampling time compared to the Alias method is approximately linear in the distribution size. For energy efficiency, it ranges from 1 to below $10^3$, indicating that the energy efficiency gains of our algorithm outweigh the increased preprocessing overhead already for small sampling sizes, even for large distributions.

**Bit efficiency compared to SOTA algorithms** Sampling algorithms are commonly evaluated based on the average number of independent fair coin flips (i.e., i.i.d. $\mathrm{Bernoulli}(0.5)$ bits) required to generate a single sample. Generating a single sample with our cLUT method requires $c$ random bits to generate the column index $J \in \{1, \dots, 2^c\}$ (i.e., uniformly sampling one of the $2^c$ entries in a row, cmp. Line 9 in Algorithm 1) and between 1 and $r$ bits to generate the row index $I \in \{1, ..., r+1\}$ (cmp. Lines 2-8 in Algorithm 1). Since $I$ follows a truncated geometric distribution, the expected number of random bits required to generate the row index $I$ is $\sum_{i=1}^{r} i \cdot 2^{-i} + r \cdot 2^{-r} = 2 - 2^{-(r-1)}$. Hence, the expected number of required random bits to produce a single sample is $b - r + 2 - 2^{-(r-1)}$.

Furthermore, $50\%$ of the generated samples consume as few as $b - r + 1$ random bits, as in these cases only a single bit is needed to generate the row index. Empirical evaluations indicate that our method is close to the information-theoretic minimal cost of sampling ($-\sum_i p_i \cdot \log_2(p_i)$, see Knuth and Yao (1976)) and approaches the minimum for high-entropy distributions (see Figure 4).

**Typical values** Discretizing (using the *finite tail extension* as detailed in the Appendix, Section B) a standard gaussian distribution to the values of the 16-bit floating point format at a precision of $b = 20$ bits (removing values with probability less than $2^{-20}$) yields $n = 20136$ values with non-zero probability, covering $99.66\%$ of the total probability mass. Applying cLUT yields a compressed table with $r = 6$, $c = 14$, and $114688$ entries ($229.38$ kB, $\rho = 9.14\times$ smaller then the uncompressed

Table 3: Average energy consumption and wall time for TrueSkill with different sampling methods.

| Method | mcp (J) | rapl:cores (J) | rapl:pkg (J) | Sampling time (s) |
|---|---|---|---|---|
| NumPy's discrete sampler | $201.05 \pm 2.45$ | $91.19 \pm 1.71$ | $116.31 \pm 1.91$ | $1.65 \pm 0.01$ |
| NumPy's continuous sampler | $160.26 \pm 1.89$ | $72.65 \pm 1.34$ | $93.36 \pm 1.71$ | $0.88 \pm 0.03$ |
| cLUT (ours) | $\mathbf{132.69 \pm 1.13}$ | $\mathbf{60.82 \pm 0.72}$ | $\mathbf{77.99 \pm 0.91}$ | $\mathbf{0.46 \pm 0.01}$ |

table). Discretizing a Gamma distribution with parameter $k = 2$ to the 16-bit format at a precision of $b = 24$ bits yields $n = 11058$ values with non-zero probability, covering $99.99\%$ of the total probability mass. Applying cLUT yields a compressed table with $r = 11$, $c = 13$, and $98304$ entries ($196.61$ kB, $\rho = 170.67\times$ smaller then the uncompressed table). Values for other precisions and distributions are shown in Figure 6 in the Appendix, Section B.

### 4.1 SAMPLING OF UNIFORM FLOATING-POINTS

Besides the proposed cLUT method, our index-based sampling scheme is ideally suited for generating uniformly distributed floating-point numbers over fixed intervals, such as the unit interval $[0, 1]$. Specifically, by considering their binary expansions, we can interpret the row and column indices generated by our method as the exponent and mantissa of the floating-point representation, respectively. Using this approach, we achieve truly uniform sampling with maximal coverage of representable values. In contrast, classic approaches for generating random numbers in fixed intervals cover only a small fraction of all representable numbers in the intervals and oftentimes fail statistical tests on uniformity (see Appendix K).

### 4.2 EXEMPLARY APPLICATIONS

In addition to evaluating the algorithm, we aim to show the potential impact of our approach on real machine learning applications. To reduce overhead and avoid confounding factors, we select a task in which sampling accounts for a significant share of total energy consumption. One example of such a task is sampling Bayesian posteriors with non-conjugate priors, and TrueSkill (Herbrich et al., 2006) system serves as an illustrative case.

The purpose of TrueSkill is to infer posterior skill distributions of players from match outcomes; this probabilistic machine learning systems currently in use on a large scale. Although the original algorithm is limited to closed-form solutions for Gaussian priors, we extend its applicability to arbitrary prior distributions through an importance sampling scheme, as detailed in the Appendix, Section H. This extension enables more flexible modeling of assumptions about the skill distributions, allowing for non-conjugate priors.

First, we conduct experiments against a fair discrete competitor (`RandomGenerator.choice` from NumPy). We then highlight the broader applicability of the approach by testing it against a fast distribution-specific sampler for a Gaussian mixture (`RandomGenerator.normal` with mixture logic from NumPy), showing that our method is effective not only for discrete unparameterized distributions but also for parametrized distributions that are (slightly) more complex than standard ones. We measure the end-to-end energy demand in the setup detailed in the Evaluation section, recording core and pkg RAPL domains. As an additional ground truth and better electricity bill proxy, we include the laptop's wall socket energy consumption using a Microchip MCP39F511N device. As a result, our method reduces the total execution time of TrueSkill by $72\%$ and decreases the total energy consumption by $34\%$ compared to the discrete sampler. Even against the specialized mixture sampler, cLUT demonstrates competitive performance with a $48\%$ reduction in sampling time and a $17\%$ decrease in overall energy consumption, as shown in Table 4. At the same time, cLUT outputs near identical posterior distribution as the two NumPy-based methods (see the Appendix H).

Additionally, an exemplary application of cLUT to the training and inference of a diffusion model is given in the Appendix, Section I.

## 5 CONCLUSION

We present cLUT, a new fast and energy-efficient sampling method for sampling from arbitrary distributions, based on operations with compressed lookup tables. Time to sample a distribution speeds up 10-100$\times$ compared to commonly used machine learning Python libraries. It saves up to 50% in energy compared to state-of-the-art methods. We further showcase the value of our sampler in real-world applications by reducing up to $34\%$ energy consumption and $72\%$ execution time in the TrueSkill example.

We provide a fairly optimized, robust, and understandable reference implementation of our algorithm in C, as well as a wrapper library that can be used with other programming languages, such as Python. We have not vectorized or parallelized our implementation to improve understandability and facilitate comparison with other methods. However, our sampling method only requires a single index-based memory lookup and some arithmetic and bit-shift operations. This makes it better suited than competing approaches for single instruction, multiple data devices (Flynn, 1972), such as modern vector and graphics processing units, given a compatible streaming source of entropy.

### ACKNOWLEDGMENTS

This research has been conducted through the funding of a research scholarship by the Hasso Plattner Foundation and was partially funded by the German Research Foundation (DFG) – 502228341 ("Memento").

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

# A NOTATION

Table 4: Notation.

| Symbol | Description |
|--------|-------------|
| $b$ | Precision of frequencies $\mathbf{f}$ in bits (e.g., minimal probability is $2^{-b}$) |
| $2^c$ | Number of columns in the compressed lookup table |
| $f$ | A vector of frequencies $\mathbf{f} = (f_1, \ldots, f_n) \in \mathbb{N}_{\geq 0}^n$ corresponding to $\mathbf{p}$ and $b$ |
| $I$ | Row index in cLUT sampling |
| $J$ | Column index in cLUT sampling |
| $N$ | Size of naive lookup table |
| $n$ | Distribution size (number of outcomes) |
| $\mathbf{p}$ | A vector of probabilities $\mathbf{p} = (p_1, \ldots, p_n) \in [0, 1]^n$ specifying the target distribution |
| $H(\mathbf{p})$ | Shannon entropy of $\mathbf{p}$ specifying the target distribution |
| $r+1$ | Number of rows in the compressed lookup table |
| $\rho$ | Compression ratio (size of compressed lookup table divided by size of naive table) |
| $\mathcal{X}$ | Domain of sampled values, e.g. the set of representable floating point numbers |
| $\mathbf{x}$ | A vector of values $\mathbf{x} = (x_1, \ldots, x_n) \in \mathcal{X}^n$ specifying the target distribution |

# B  DETAILS ON NON-FINITE DISTRIBUTIONS

Many distributions relevant to machine learning belong to the class of continuous, real-valued, uni-variate distributions, with the Gaussian distribution as a prominent example. These distributions are discretized in a computational setting, as hardware can only represent a finite set of values.

A natural discretization proceeds as follows. Let a distribution on $\mathbb{R}$ be specified via its cumulative density function $F$. To discretize it on a finite support $\mathcal{X} \subset \mathbb{R}$ ($|\mathcal{X}| < \infty$), e.g., the set of representable numbers in the IEEE 754 16-bit floating-point format, we define the probability mass function $p : \mathcal{X} \to [0, 1]$ of the discretized distribution by

$$p(x) := \frac{1}{c}\Big[F\Big(\frac{x + x_+}{2}\Big) - F\Big(\frac{x + x_-}{2}\Big)\Big], \quad \forall x \in \mathcal{X},$$

where $x_+ := \min\{y \in \mathcal{X} : y > x\}$ is the next number to the right of $x$ in $\mathcal{X}$, and $x_- := \max\{y \in \mathcal{X} : y < x\}$ is the next number to the left.

Special care is required for the extrema of $\mathcal{X}$. Let $x^{\max} := \max \mathcal{X}$ and $x^{\min} := \min \mathcal{X}$. The next numbers beyond these limits can be defined in two ways, depending on how you would like to attribute the probability mass of the tails:

1. *Finite tail extension:*

$$x_+^{\max} := x^{\max} + \frac{x^{\max} - x_-^{\max}}{2},$$
$$x_-^{\min} := x^{\min} - \frac{x_+^{\min} - x^{\min}}{2},$$

   which requires a normalization constant $c = 1 - F(x_+^{\max}) + F(x_-^{\min})$ to ensure that the discretized probability mass function sums to one.

2. *Infinite tail extension:* $x_+^{\max} := +\infty$ and $x_-^{\min} := -\infty$, in which case $c = 1$ suffices.

Discrete distributions with infinite support, such as the Poisson distribution over $\mathbb{N}_{\geq 0}$, also require truncation to be represented in finite precision. A common approach is to apply a cutoff.

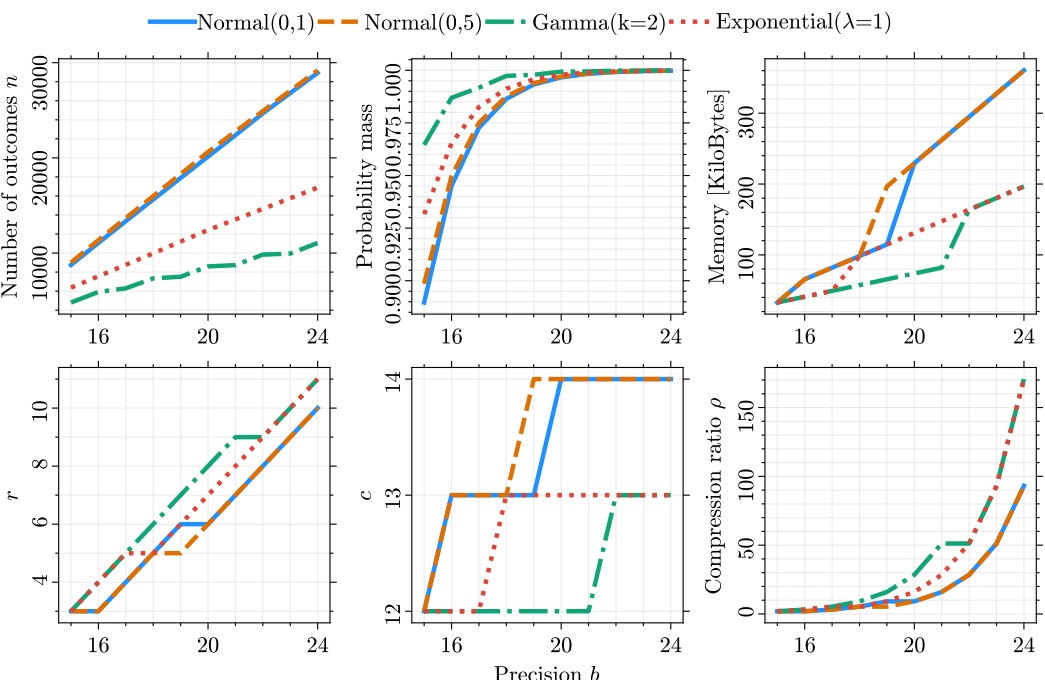

Figure 6: Typical values for classic continuous distributions when discretized to the 16-bit floating point range $\mathcal{X}$ with a precision of $b \in \{15, \ldots, 24\}$, as described in section 3. Shown are (1) the number of outcomes $n$, i.e., the number of values with non-zero probability, and (2) the covered probability mass (sum of all probabilities before normalizing) after rounding to precision $b$. (3) Memory consumption, (4) row parameter $r$, (5) column parameter $c$, and (6) achieved compression ratio $\rho$ of the compressed lookup table.

## C  DETAILS ON APPROXIMATED DISTRIBUTIONS

Since memory constrains impose a boundary on the size $N$ of any lookup table, a lookup table might suffer from the inability to represent certain probabilities, such as very small or irrational probabilities, e.g., $p_i = \sqrt{1/2}$. In these cases we fill the table according to the frequencies

$$f_i := \text{round}(p_i \cdot 2^b) \in \mathbb{N}_{\geq 0}, \quad i \in \{1, \ldots, n\},$$

where $\text{round}(\cdot)$ is an arbitrary sum-preserving rounding scheme. The approximation error of a distribution stored in a lookup table with probabilities $f_i/2^b$ directly depends on the precision $b$, as an upper bound on the KL divergence can be expressed as a function of $\min_{1 \leq i \leq n} f_i$ (see Theorem 1).

**Theorem 1** (KL-Divergence of approximated distribution). *The KL-Divergence between a distribution on $\mathbf{x} = (x_1 \ldots, x_n)$ given by the associated probabilities $\mathbf{p} = (p_1, \ldots, p_n)$ and the distribution approximated to a precision of $b \in \mathbb{N}_{>0}$ bits given by the frequencies $\mathbf{f} = (f_1, \ldots, f_n)$ is bounded by*

$$D_{KL}(\mathbf{p} \,||\, \mathbf{f}) \leq \log\left(1 + \frac{1}{2\kappa}\right),$$

*where $\kappa := \min_{1 \leq i \leq n} f_i$.*

*Proof.* Write

$$p_i = f_i \cdot 2^{-b} + \delta_i,$$

with $\delta_i \in [-2^{-b-1}, 2^{-b-1}]$. Then, the KL-Divergence is given by

$$\begin{aligned}
D_{\text{KL}}(\mathbf{p} \,||\, \mathbf{f}) &= \sum_{i=1}^{n} p_i \log \frac{p_i}{f_i \cdot 2^{-b}} \\
&= \sum_{i=1}^{n} p_i \log \frac{f_i \cdot 2^{-b} + \delta_i}{f_i \cdot 2^{-b}} \\
&= \sum_{i=1}^{n} p_i \log \left(1 + \frac{\delta_i \cdot 2^b}{f_i}\right) \\
&\leq \sum_{i=1}^{n} p_i \log \left(1 + \frac{2^{-b-1} \cdot 2^b}{\min_i f_i}\right) \\
&= \log \left(1 + \frac{1}{2 \min_i f_i}\right) \cdot \sum_{i=1}^{n} p_i \\
&= \log \left(1 + \frac{1}{2\kappa}\right),
\end{aligned}$$

where $\kappa := \min_{1 \leq i \leq n} f_i$. In the third step, we used that $\delta_i \leq 2^{-b-1}$ and $f_i > \min_i f_i$ for all $i$. $\qquad\square$

Note that $\kappa = \min_{1 \leq i \leq n} f_i = \min_{1 \leq i \leq n} \text{round}(p_i \cdot 2^b)$ and therefore $D_{\text{KL}} \in \mathcal{O}(\log(1 + 2^{-b}))$. Clearly, $D_{\text{KL}} \to 0$ for $b \to \infty$. However, while approximation error decreases logarithmically with precision $b$, the lookup table size $N$ required to store all values $\mathbf{x}$ with their respective frequencies $\mathbf{f} = (f_1, \ldots, f_n)$ grows exponentially in $b$:

$$N := \sum_{i=1}^{n} f_i = 2^b.$$

## D    PREPROCESSING DETAILS

A pseudo code of the cLUT preprocessing algorithm that constructs the compressed lookup table is shown in Algorithm 2. Algorithm 2 calls a the function `distribute()` in line 4, which is detailed in pseudo code in Algorithm 3.

---

**Algorithm 2** Constructing a compressed lookup table

---

**Require:** probability distribution given by $\mathbf{x} = (x_1, x_2, \ldots, x_n)$ and $\mathbf{f} = (f_1, f_2, \ldots, f_n) \in \mathbb{N}_{\geq 0}^n$
**Ensure:** compressed lookup table `compressedTable` of size $(r + 1) \times 2^c$
    ▷ *Compute optimal r and c:*
1: $b \leftarrow \log_2(\sum_{i=1}^n f_i)$
2: $r \leftarrow \max\{v \in [0, b] : \sum_{j=0}^w \sum_{i=1}^n f_i^{(j)} \cdot 2^{v-b-1} \leq 1 \quad \forall w \in \{0, \ldots, b\}\}$
3: $c \leftarrow b - r$
    ▷ *Compute counts per row for each value:*
4: $D \leftarrow \texttt{distribute}(\mathbf{f}, r, c)$
    ▷ *Fill compressed lookup table:*
5: `compressedTable` $\leftarrow [\,]$
6: **for** $i = 1$ to $r+1$ **do**
7:     **for** $j = 1$ to $n$ **do**
8:         **for** $k = 1$ to $D_{ji}$ **do**
9:             `compressedTable`.append($x_j$)
10:         **end for**
11:     **end for**
12: **end for**
13: **return** `compressedTable`

---

**Algorithm 3** Distribute counts across bit levels with `distribute()`

---

**Require:** frequencies $\mathbf{f} = (f_1, f_2, \ldots, f_n) \in \mathbb{N}_{\geq 0}^n$, $r \in \mathbb{N}_{\geq 0}^n$, $c \in \mathbb{N}_{\geq 0}^n$
**Ensure:** bit levels $D \in \mathbb{N}_{\geq 0}^{n \times r+1}$
    ▷ *Expand counts into bit-level representation*
1: **for** $i = 1$ to $n$ **do**
2:     **for** $j = 1$ to $b$ **do**
3:         $D_{ij} \leftarrow f_i^{(j)}$
4:     **end for**
5: **end for**
    ▷ *Redistribute bits above level r*
6: **for** $k = b$ downto $r$ **do**
7:     **for** $i = 1$ to $n$ **do**
8:         $D_{ir} \leftarrow D_{ir} + 2^{k-r+1} \cdot D_{ik}$
9:     **end for**
10: **end for**
    ▷ *Adjust lower levels if cumulative sum exceeds $2^c$*
11: **for** $k = r - 1$ downto $1$ **do**
12:     $a \leftarrow 0$
13:     **for** $i = 1$ to $n$ **do**
14:         $a \leftarrow a + D_{ik}$
15:         **if** $a > 2^c$ **then**
16:             $\delta \leftarrow a - 2^c$
17:             $D_{ik} \leftarrow D_{ik} - \delta$
18:             $D_{i(k-1)} \leftarrow D_{i(k-1)} + 2 \cdot \delta$
19:         **end if**
20:     **end for**
21: **end for**
22: **return** $(D_{ij})_{j \leq r+1}$

---

## E  IMPLEMENTATION DETAILS

We implemented the preprocessing and sampling methods in C and reused the computed data structures in Python. To do so, we created a foreign function library that conveniently interfaces between C and other languages. This library is used in our evaluation.

Like the reference implementation of ALDR and FLDR (Draper and Saad, 2025), we used bit operations, compiler intrinsics and linearized arrays where possible to ensure fast computation. We extended the existing SOTA implementations to also work with 64-bit input values to make them comparable with our test distributions.

Our implementation, wrapper library and changes to existing SOTA implementations are publicly available on GitHub under (omitted for blind review).

# F  JAX INTEGRATION AND GPU EVALUATION

To demonstrate the integratability of cLUT as well as potential performance gains from SIMD implementations, we have integrated our cLUT approach into the JAX library, as shown in Listing 1, and compared with the default sampling method from JAX, see Figure 7. This experiment was run on a single A100 GPU, using JAXs internal GPU mechanisms.

Listing 1: Integration of cLUT into the JAX library.

```python
# jax._src.random.py
from jax._src import numpy as jnp
from jax._src import prng
...

def choice(key: ArrayLike,
           a: int | ArrayLike,
           shape: Shape = (),
           replace: bool = True,
           p: RealArray | None = None,
           # ---- CHANGES ----
           b = -1,
           c = -1,
           # ---- END OF CHANGES ----
           axis: int = 0,
           mode: str | None = None) -> Array:
    ...
    if replace:
        # ---- CHANGES ----
        return _choice(arr, key, c, b, shape, dtype)
        # ---- END OF CHANGES ----
    else:
        ...

# ---- CHANGES ----
@partial(jax.jit, static_argnames=['b', 'shape', 'dtype'])
def _choice(arr, key, c, b, shape, dtype):
    mask = (1 << c) - 1
    B = prng.random_bits(key, b, shape)
    return jnp.take(arr, (clz(B | mask) << c) | (B & mask), 0)
# ---- END OF CHANGES ----
```

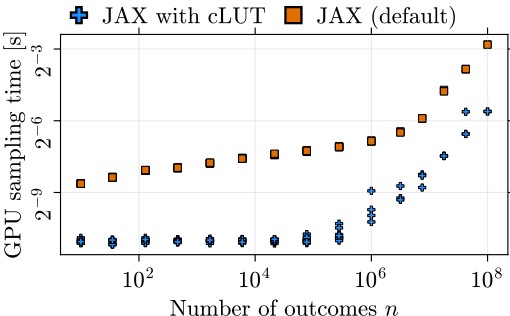

Figure 7: Comparison of our cLUT approach integrated into the JAX library with the default sampling method from JAX on GPU. Shown is the average wall time (in seconds) to generate $10^7$ samples from distributions of varying sizes $n \in [10^1, 10^8]$. Distributions were extracted from exponential distributions with varying parameters (and shuffled) to cover a broad range of entropies, using variable precisions $b \in [4, 30]$. The plots are shown on a log-log scale. Each measurement was repeated ten times and averaged.

# G ADDITIONAL EVALUATIONS

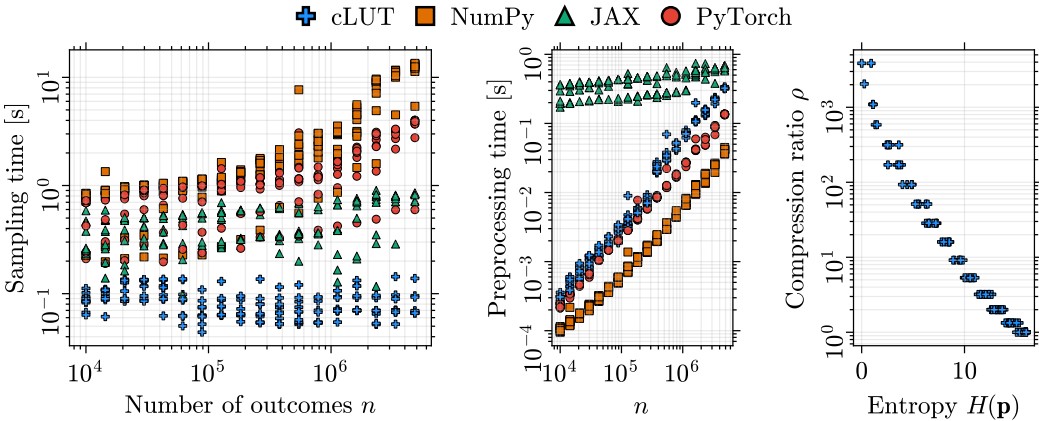

Figure 8: Comparison of our cLUT approach with standard sampling methods from popular machine learning libraries (NumPy, JAX, PyTorch). Similar to Figure 3, but evaluated on sparse distributions. Shown are (1) the average wall time (in seconds) to generate $10^7$ samples from distributions of varying sizes $n \in [10^4, 10^7]$, (2) the preprocessing time, and (3) the compression ratios $\rho$ of the cLUT algorithm. Distributions were sampled from Dirichlet priors with varying parameters to cover a broad range of entropies, using a fixed precision of $b = 16$. The plots are shown on a log-log scale. Each measurement was repeated five times and averaged.

Table 5: Average wall time (in seconds, mean $\pm$ std) for generating $10^7$ samples and the preprocessing step. Evaluated in two subsets of the distributions from Figure 8, split by size.

| # Outcomes: | $n \in [10^4, 10^5]$ | | $n \in [10^6, 10^7]$ | |
|---|---|---|---|---|
| **Method** | **Sampling time (s)** | **Preprocessing time (s)** | **Sampling time (s)** | **Preprocessing time (s)** |
| NumPy | $0.847 \pm 0.288$ | $\mathbf{0.000 \pm 0.000}$ | $7.230 \pm 4.138$ | $\mathbf{0.020 \pm 0.012}$ |
| PyTorch | $0.720 \pm 0.253$ | $0.001 \pm 0.001$ | $2.400 \pm 1.138$ | $0.070 \pm 0.039$ |
| JAX | $0.407 \pm 0.132$ | $0.335 \pm 0.087$ | $0.616 \pm 0.250$ | $0.572 \pm 0.105$ |
| cLUT (ours) | $\mathbf{0.095 \pm 0.025}$ | $0.001 \pm 0.001$ | $\mathbf{0.080 \pm 0.022}$ | $0.177 \pm 0.090$ |

Table 6: Average energy demand, wall time, and power draw of a single sampling operation. The power draw series is computed by dividing the energy series by wall time. It averages over all CPU instructions of a sampling iteration. High variance in entropy and distribution size results in the high standard deviation here. Shown are distributions from Figure 9, split by size.

| # Outcomes: | $n \in [10^4, 10^5]$ | | | $n \in [10^6, 10^8]$ | | |
|---|---|---|---|---|---|---|
| **Method** | **Energy (nJ)** | **Time (ns)** | **Power (W)** | **Energy (nJ)** | **Time (ns)** | **Power (W)** |
| ALDR | $240.65 \pm 73.91$ | $18.83 \pm 4.75$ | $13.21 \pm 4.28$ | $225.38 \pm 53.64$ | $19.56 \pm 4.56$ | $12.29 \pm 4.61$ |
| FLDR | $221.25 \pm 62.66$ | $20.08 \pm 4.99$ | $11.45 \pm 3.75$ | $204.54 \pm 47.07$ | $20.83 \pm 4.14$ | $10.24 \pm 3.56$ |
| Alias | $180.58 \pm 73.82$ | $17.94 \pm 7.82$ | $\mathbf{10.21 \pm 0.87}$ | $315.04 \pm 93.36$ | $33.89 \pm 9.94$ | $\mathbf{9.32 \pm 0.48}$ |
| cLUT (ours) | $\mathbf{144.09 \pm 18.34}$ | $\mathbf{14.15 \pm 1.58}$ | $10.22 \pm 1.06$ | $\mathbf{128.23 \pm 17.05}$ | $\mathbf{13.41 \pm 1.75}$ | $9.60 \pm 0.94$ |

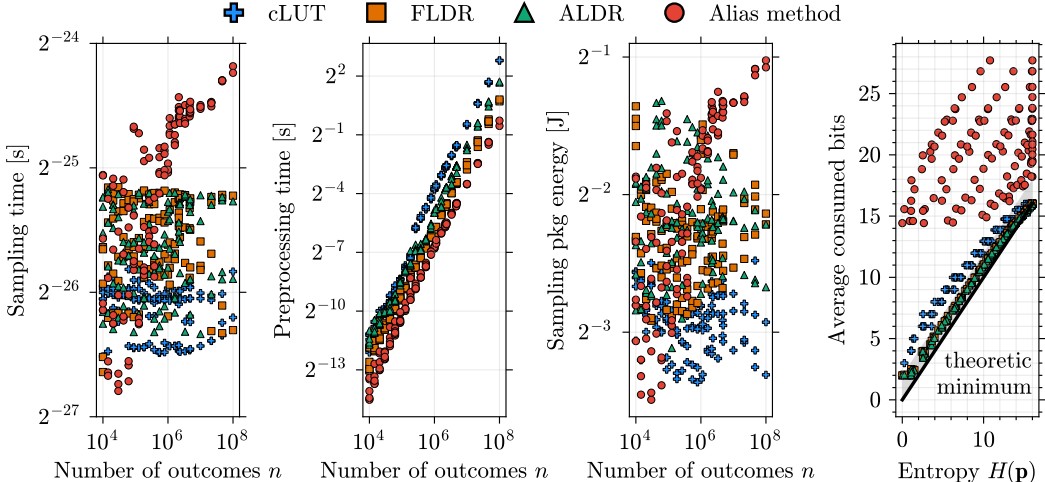

Figure 9: Comparison of our cLUT approach with existing state-of-the-art sampling methods. Similar to Figure 4, but evaluated on sparse distributions described in Figure 8. Shown are (1) the wall time required for generating a single sample (averaged over $10^6$ repetitions) and (2) preprocessing (averaged over 10 repetitions), as well as (3) the cumulative energy demand of the CPU socket for generating $10^6$ samples. Time and energy are shown on a log-log scale. The fourth subfigure shows the average consumed bits per sample from the entropy source.

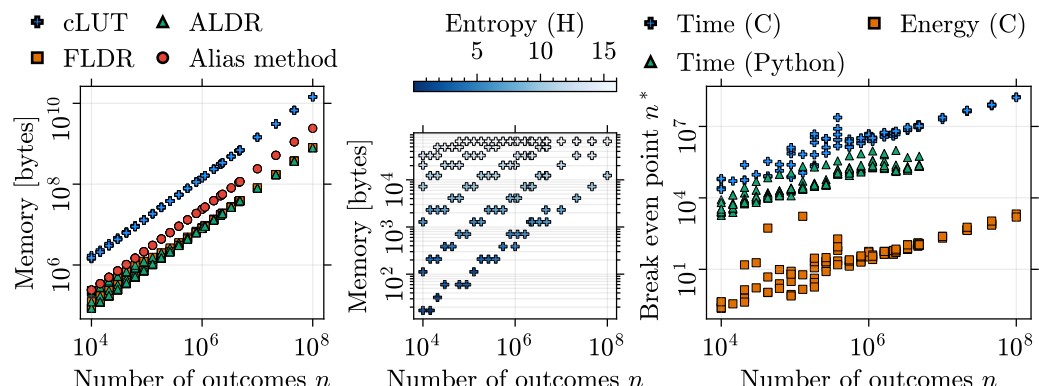

Figure 10: Comparison of cLUT with state-of-the-art sampling methods in terms of memory usage and break-even analysis. Similar to Figure 5, but evaluated on sparse distributions described in Figure 8. (1) peak memory usage for all methods (including preprocessing), (2) memory usage of compressed cLUT table, and (3) break-even analysis against the Alias method. The break-even point $n^*$ is the minimum number of samples needed for cLUT to offset its preprocessing overhead relative to the Alias method (in terms of sampling time or energy consumption).

## H    DETAILS ON TRUESKILL

Our TrueSkill extension uses importance sampling as follows: (1) independently sample skills $s_i$ and performances $y_i$ from their respective priors, (2) compute importance weights as the product of prior densities and match likelihood, and (3) use these weights to estimate posterior distributions. Independent sampling of correlated variables enables parallelization while maintaining correctness through importance weighting (Algorithm 4). We discretize the continuous bimodal prior over the range $[-10, 10]$ with resolution $10^{-3}$ and construct cLUT tables with $b = 32$ bit precision.

To evaluate the precision of the posterior distribution sampled by cLUT, we ran the TrueSkill algorithm 50 times using both the NumPy-based continuous sampler and the cLUT sampler. Considering

---

**Algorithm 4** TrueSkill with importance sampling for two players

---

**Require:** prior skills distributions $\pi_1(\theta_1)$ and $\pi_2(\theta_2)$, performance standard deviation $\beta$, match outcome data $R$
**Ensure:** posterior skills distributions $\pi_1(\theta_1|R)$ and $\pi_2(\theta_2|R)$
 1: **for** $i = 1$ **to** $N$ **do**
 2:      $s_1 \leftarrow \pi_1(\theta_1), s_2 \leftarrow \pi_2(\theta_2)$
 3:      $y_1 \leftarrow \mathcal{G}(1, \beta), y_2 \leftarrow \mathcal{G}(1, \beta)$
      ▷ *Compute match outcome:*
 4:      $r = \mathbb{I}_{y_1 > y_2}$
      ▷ *Compute importance sampling weights:*
 5:      $w_1 = p_{\pi_1|\theta_1}(s_1), w_2 = p_{\pi_2|\theta_2}(s_2)$
 6:      $w_3 = p_{\mathcal{G}(s_1, \beta)}(y_1), w_4 = p_{\mathcal{G}(s_2, \beta)}(y_2)$
 7:      $w = r \cdot \prod_{i=1}^{4} w_i$
      ▷ *Write down the results to arrays $S_1, S_2, W$:*
 8:      $S_1[i] = s_1, S_2[i] = s_2, W[i] = w$
 9: **end for**
      ▷ *Assign new posterior distribution as probability mass function:*
10: $\pi_1(\theta_1)|R) := \{(S_1[i], W[i])\}_{i=1}^{N}$
11: $\pi_2(\theta_2|R) := \{(S_2[i], W[i])\}_{i=1}^{N}$
12: **return** $\pi_1(\theta_1|R), \pi_2(\theta_2|R)$

---

that these two samples operate on different domains, we cannot employ test that compare density functions. For this reason, we evaluate sampled results by comparing first and second moments. For each iteration, we computed the mean and variance of a player's skill posterior distribution. We then applied a t-test to assess statistically significant differences in means and variances between the two samplers, obtaining p-values greater than 0.2 in both cases, meaning that the moments of sampled distributions do not have meaningful differences.

## I   APPLICATION TO DIFFUSION MODELS

To demonstrate cLUT's impact in a core ML problem, we apply cLUT to a small-scale generative model. We train and validate a toy diffusion model Ho et al. (2020a) designed to learn a noise distribution from corrupted data. In our experiments, we generate the training data from a bimodal distribution (green line in the Figure 11) and introduce corruption through another bimodal distribution. While the original algorithm assumes training and inference with Gaussian noise, previous work has shown that reducing the difference between the data and noise distributions can improve the precision of a model Lee et al. (2021). Additionally, using a Gaussian mixture can be a beneficial replacement for certain tasks Nachmani et al. (2021). Our additional experiments are consistent with these findings: when training on bimodal data, using Gaussian noise results in a substantially larger Wasserstein distance between generated samples and the training distribution (greater than 1), while using bimodal noise reduces this distance to below 0.07.

Sampling is employed to simulate noise during both training and inference. We incorporate cLUT in both stages and compare its performance and energy consumption against the default sampling in JAX, as JAX was the most efficient library in our main evaluation. For the CPU evaluation, we use the same hardware setup as described before. We define a shallow neural network with two linear layers and train on small batches of 8 samples for $3 \times 10^5$ steps. For the inference stage, we run the trained model for $2 \times 10^3$ iterations with the same batch size. To sample noise with the cLUT algorithm, we construct a table with a fixed precision of $b = 8$. This preprocessing cost is included in the evaluation of the overall application's time and energy consumption.

Table 7 shows that incorporating cLUT can save energy by $37\%$ in the training stage and by $65\%$ in the inference stage compared to the default sampler of JAX. Additionally, to validate the quality of generated samples, we compare the output of the inference stages using the two different sampling algorithms, utilizing a model trained with JAX's default sampler. As shown in Figure 11, the two samplers return nearly identical distributions for the generated data, with a Wasserstein distance from training data to samples of 0.069 for JAX's default and 0.054 for cLUT, respectively.

Table 7: Comparison of our cLUT approach with JAX incorporated into training and inference processes of a denoising diffusion model.

| | Training | | | Inference | | |
|---|---|---|---|---|---|---|
| **Method** | **rapl:cores (J)** | **rapl:pkg (J)** | **Time (s)** | **rapl:cores (J)** | **rapl:pkg (J)** | **Time (s)** |
| JAX | $3811.42 \pm 102.38$ | $4702.91 \pm 86.61$ | $265.77 \pm 5.29$ | $331.63 \pm 3.67$ | $403.29 \pm 6.53$ | $19.89 \pm 0.22$ |
| cLUT | $2392.14 \pm 111.05$ | $2962.17 \pm 102.96$ | $172.62 \pm 5.11$ | $116.11 \pm 1.97$ | $143.08 \pm 3.06$ | $7.28 \pm 0.08$ |
| Reduction with cLUT | 37.2% | 37.0% | 35.0% | 65.0% | 64.5% | 63.4% |

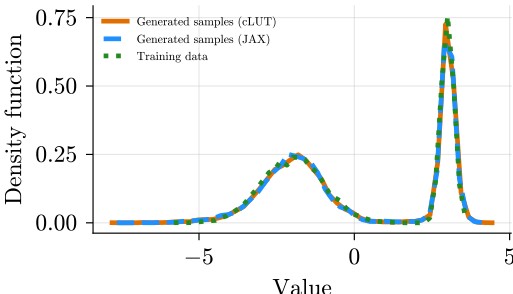

Figure 11: Generated data by the denoising diffusion model with cLUT and JAX sampling algorithms and comparison to the target data.

## J  DETAILS ON ENERGY EFFICIENCY

It is crucial to understand different metrics and their relation to assess the efficiency of modern (electrical) computing systems and design experiments. While power is the rate at which electricity is consumed at a given point in time, energy is the amount of electricity required to perform an operation (power's integral over time). Electric energy translates to battery life, electricity bills or emitted carbon dioxide, making it the most reasonable metric to optimize for when seeking *energy efficiency*.

An exception would be if the computer system has actively changing clock frequencies. Apart from the number of active switching transistors, the CPU's clocking frequency and supply voltage play into the dynamic power demand at a given point in time (Le Sueur and Heiser, 2010). In this case, the energy-delay-squared product (Martin et al., 2002) would be a more suitable metric, combining execution time and energy demand.

Even at fixed clock rates, switching between CPU architectures can significantly alter power demand but not necessarily energy demand. A low-power device (a micro-controller or efficiency CPU core) can run for a longer time than a more power-intense one, resulting in comparable energy integrals—or not, depending on the static power demand and thus energy proportionality of the system (Barroso and Hölzle, 2007). For a fixed problem size, the latter device can switch to idle mode after completion or process more elements for a given unit of energy. Consequently, to obtain more representative measurements, we fixed the CPU frequency and micro-architecture (cores) in our experiments. As our particular Intel *Hybrid* CPU architecture comprises of larger **p**erformance cores and limited **e**fficiency cores, we opted for the P-cores for consistent measurements.

There is a direct connection between the memory access behavior of modern computer systems and their electricity consumption (Horowitz, 2014). Memory subsystems and CPU caches have long been overlooked in comparison to computational cores but constitute a large portion of active transistors in today's chip designs, leading to higher dynamic power demands. This means that, for general-purpose computers, algorithms that trade computation for memory lookups may have slightly worse energy efficiency than plain recomputation. This effect is more pronounced with multiple, nested lookups (also known as *pointer chasing*) because it involves more active transistors, which increases power demand. It also breaks CPU cache locality and access prediction, resulting in prolonged CPU stalls (increased time demand) and thus non-linear increase in energy demand. This motivates our idea to create a compression strategy for a lookup table that preserves all the statistical properties of sampling with simple lookup tables but reduces energy consumption.

## K  DETAILS ON SAMPLING OF UNIFORM FLOATING-POINTS

In the IEEE 754 floating-point format, numbers are organized into dyadic intervals of exponentially increasing size, each containing a fixed number of equally spaced values. This structure makes our index-based sampling scheme ideally suited for generating uniformly distributed floating-point numbers over fixed intervals, such as the unit interval $[0, 1]$. Specifically, by considering their binary expansions, we can interpret the row and column indices generated by our method as the exponent and mantissa of the floating-point representation, respectively. Using this approach, we achieve truly uniform sampling with maximal coverage of representable values.

In contrast, the classic approach of generating uniformly random mantissa bits to obtain a float in $[1, 2)$, and then subtracting 1 covers only a small fraction of all representable numbers, approximately 13%. PyTorch's common method for generating random variates uniformly on the interval $[0, 1]$ is `torch.rand()`. When generating values directly in 16-bit floating-point format, this method covers only 13.3% of all representable values in $[0, 1]$. A Pearson's $\chi^2$ test for uniformity fails significantly, yielding $\chi^2 = 1{,}277{,}749{,}854.249$ with $p < 10^{-10}$. Alternatively, generating values in 32-bit floating-point format and converting them to a 16-bit representation results in 100% coverage of 16-bit floating-point values in the unit interval. However, this approach also fails the Pearson's $\chi^2$ test, with $\chi^2 = 21{,}425.2924$ and $p < 10^{-10}$.

