# OpenReview forum: "Energy-Efficient Random Variate Generation via Compressed Lookup Tables"
_ICLR.cc/2026/Conference — ICLR 2026 Poster_

### Official Review · Reviewer_QjRf · 2025-10-27

**Soundness:** 3
**Presentation:** 4
**Contribution:** 3
**Rating:** 8
**Confidence:** 1

**Summary:**

This paper proposes a new sampling strategy for random variate generation from arbitrary distributions. The core idea is to construct a lossless compressed lookup table and perform efficient sampling from it. The approach to generating the compressed lookup table appears to be computationally efficient and potentially innovative.

**Strengths:**

The paper is well-written, and the experiments are sound. The baselines are well-chosen, ranging from generic Python library routines commonly used in the machine learning community to state-of-the-art competing methods for this particular task. The paper’s emphasis on energy efficiency is commendable and highlights an often-overlooked but important aspect of algorithmic design.

**Weaknesses:**

The absence of a computation complexity analysis appears to be the paper’s main weakness, though the authors may have a valid reason for omitting it.

**Questions:**

__Q1.__ In the conclusion section, you mention potential advantages of implementing your method on GPU. Since one of the comparisons is against GPU-accelerated libraries while your experiments are conducted only on the CPU, could you elaborate on these points in more detail? specifically, how these advantages arise compared to other methods and how they could be effectively leveraged or exploited in practice?

__Q2.__ While many discrete sampling routines in Python packages use C++ backends, how do you ensure that Python wrapper overhead does not skew your runtime comparisons?

__Q3.__ In Table 1, why is the sampling time faster for $n \in [10^6, 10^7]$ in your proposed method compared to smaller $n$ ($n \in [10^4, 10^5]$), while the other methods have slower sampling times when $n$ is larger?

__Q4.__ It might be helpful to also show, in Figure 5, the peak memory usage excluding the preprocessing cases. Additionally, it would be nice to include other memory-related metric(s) such as average memory usage as well as the overall memory behavior during execution.

__Minor__

- In line 107, the real RAM assumption should be introduced more clearly, perhaps with one or two explanatory sentences or a short note in the appendix. Instead of only referring to Shamos (1978). At the minimum, please provide the abbreviation of RAM (Random Access Machine).
- In line 149, while $c$ and $b$ are quite self-explanatory, it might be helpful to add a brief sentence explaining what $r$ represents to give readers a quick intuition. In general, a short description of all notation used could greatly improve the clarity of the paper.

---

> ### Author Response · Authors · 2025-11-20
>
> We thank the reviewer for their positive assessment, constructive questions, and helpful suggestions on clarity.
>
> - **On weaknesses**:
> We thank the reviewer for raising this point. To better address it in the revision, we would appreciate clarification on what specific aspects of the complexity the reviewer had in mind.
> Any guidance on which part the reviewer felt was missing or most relevant would help us strengthen the corresponding section of the paper.
>
> - **On questions:**
> We agree with the reviewer that the effect of the Python wrappers in the libraries we compare against is not to be neglected. In order to minimize the potential effect, we split the runtime analysis in preprocessing time and sampling time. Furthermore, we woudl like to note that we did not use our C implementation directly in this evaluation, but rather a Python wrapper for the preprocessing and a Cython implementation of cLUT, that draws it's randomness from NumPy's random uniform generator, to facilitate a fair comparison that accounts for Python overhead.
> We have added Figure 6 in the appendix that reports the memory requirements of the compressed table (essentially what needs to be kept in memory after preprocessing) of commonly used distributions for different precisions, to get a sense of typical values.
> We have added a comparison of our cLUT approach integrated into the JAX library with the default sampling method from JAX. This experiment was run on a single A100 GPU, using JAXs internal mechanisms. We have demonstrated a persistent performance acceleration. Details are in the Appendix, Section F and Figure 7.
>
> - **On minor comments:**
> We thank the reviewer for pointing these out.
> We have provided the abbreviation and a short explanatory sentence of RAM in the main body.
> Furthermore, we have added a table of notations in the Appendix, Section A, to improve the clarity of the paper.
>
> We thank the reviewer again for their constructive feedback and for highlighting improvements that have strengthened the clarity and completeness of the paper.

---

> > ### Comment · Reviewer_QjRf · 2025-11-21
> >
> > > we would appreciate clarification on what specific aspects of the complexity the reviewer had in mind. Any guidance on which part the reviewer felt was missing or most relevant would help us strengthen the corresponding section of the paper.
> >
> > By “computational complexity,” I meant the Big-O complexity in terms of n, describing how the computation grows as n increases. However, I think, the runtime experiments appear sufficient without this, so I will keep my score as it is.

---

### Official Review · Reviewer_5BMn · 2025-10-30

**Soundness:** 3
**Presentation:** 3
**Contribution:** 3
**Rating:** 6
**Confidence:** 2

**Summary:**

This paper introduces cLUT (compressed Lookup Tables), a sampling method designed to generate samples from arbitrary discrete distributions efficiently. The authors claim that sampling, a core component of many machine learning algorithms is a significant computational and energy bottleneck.

The proposed method is based on two key contributions (1) a lossless compression scheme via a constructing a compressed lookup table and (2) a fast algorithm to sample efficiently given the compressed lookup table.

The authors provide a Python and C implementation of their algorithm. The results are demonstrate that the Python implementation of cLUT is 10-100x faster than standard Python samplers (NumPy, PyTorch, JAX), depending on the number of samples. The C implementation of cLUT is more energy-efficient and faster than state-of-the-art (SOTA) C-based samplers like ALDR, FLDR, and the Alias method. However, this efficiency comes at the cost of a higher one-time preprocessing step. The method's practical benefits are demonstrated in a modified TrueSkill application, where it significantly reduces execution time and energy consumption.

**Strengths:**

**Significance**: The paper addresses a fundamental, practical, and important problem: the high energy and computational cost of sampling in ML. The claimed 33-60% energy savings and 10-100x speedup over common Python libraries is impressive.

**Originality**: I am not an expert in the particular area of low-level implementation of sampling algorithms. But the core idea of combining a specific binary-expansion-based compression with a geometric-plus-uniform index sampling scheme appears to be novel.

**Quality of evaluation**: The authors wisely compare against two separate groups: (1) high-level, commonly used Python libraries and (2) low-level, high-performance SOTA C implementations and demonstrate improved performance in terms of sampling time and energy efficiency across the board. They also will publicly release the implementation. The authors are upfront about the method's primary trade-off of cLUT's higher pre-processing time.

**Clarity**: The paper is mostly well-written and is easy to follow. I have some suggestions for further improvements that are detialed in the next sections.

**Weaknesses:**

**Missing Algorithmic Details on Preprocessing**: The preprocessing step, which constructs the compressed table, is a critical part of the contribution but is not well-explained. The main text describes a "rectification" process to ensure all rows have a uniform width, which is essential for the sampling scheme. This section refers to Algorithm 2 in the appendix. However, Algorithm 2 is itself a high-level sketch. It calls a function distribute(z, f, r, c)  which seemingly performs all the work, but this function is undefined. The visual example in Figure 2 is helpful but insufficient to understand how this redistribution is performed algorithmically.

**Vagueness on Key Parameter $r$**: The performance, compression ratio $\rho = 2^r / (r+1)$, and bit efficiency are all critically dependent on the number of rows, $r$. The paper is vague on how $r$ is determined, stating it "depend[s] on the frequencies f". The formula provided in Algorithm 2 (Line 2) 33 is complex and presented without any derivation or intuition. The paper would be much stronger if it provided bounds or illustrative examples of what $r$ typically is for common distributions (e.g., uniform, Gaussian) at a given precision.

**Limited ML Application Context**: The introduction strongly motivates the work by citing its relevance to VAEs, contrastive learning, and diffusion models. However, the evaluation only uses an extended TrueSkill system. While this is a valid real-world test, it is a relatively niche application. To compellingly demonstrate the paper's significance to the ICLR community, it would be far more powerful to show the effectiveness of plugging cLUT into even a small-scale generative model (like a VAE or a simple diffusion model) during training or inference. As is, the contribution feels more like a general-purpose algorithm paper than a paper demonstrating a clear impact on a core ML problem.

**Questions:**

I am wondering how generalizable the proposed method is. Is it possible to plug cLUT into pytorch so that all the sampling operations uses cLUT instead of whatever the default sampling algorithm is? If so, what would the end-to-end effect of such update be on a typical training/inference task?

---

> ### Author Response · Authors · 2025-11-20
>
> We thank the reviewer for their thoughtful assessment, constructive suggestions, and for highlighting both strengths and areas for clarification.
>
> - **On "Missing Algorithmic Details on Preprocessing"**:
> In the revised version, we have substantially expanded Appendix D by providing pseudocode for the previously abstracted `distribute()` function.
>
> - **On "Vagueness on Key Parameter r":**
> Following the reviewer's and reviewer PkGd's suggestion, we added a paragraph "Typical values" in the Evaluation (Section 4), and Figure 6 in the Appendix (Section B) reporting, amongst others, typical values of $r$ for common continuous distributions (Normal, Gamma, Exponential) and different precisions $b$.
>
> - **On questions:**
> cLUT could be incorporated into existing frameworks as an alternative to the discrete sampler routine (e.g. `numpy.rng.choice`, `torch.multinomial`, `jax.random.choice`). It can be integrated either (i) by directly calling the provided C wrapper or (ii) via native framework implementations. cLUT only requires access to Bernoulli random bits already provided by these libraries. To clarify this, we have added a paragraph in the Appendix demonstrating how cLUT could be integrated into JAX’s random.choice, illustrating a minimal native implementation. We emphasize that the TrueSkill experiment serves to demonstrate the end-to-end effects in a concrete instantiation of such an integration.
>
> We thank the reviewer again for the constructive suggestions. We believe the revisions have improved clarity regarding preprocessing, the behavior of key parameters, and the integration possibilities.

---

> > ### Comment · Reviewer_5BMn · 2025-11-21
> >
> > I appreciate authors' response and updates to the draft. I still think a demonstration on a modern generative model like a diffusion model's training or sampling process makes it a much stronger paper.
> >
> > Regardless, I remain supportive of the paper's acceptance.

---

> > > ### Author Response · Authors · 2025-11-27
> > >
> > > We appreciate the reviewers' response.
> > > As suggested by the reviewer, we added an experiment evaluating the exemplary application of cLUT to a small-scale generative model, demonstrating cLUT's impact in a core ML problem highly relevant to the ICLR community.
> > > Incorporating cLUT in a denoising diffusion model resulted in energy gains of 37% and 65% for training and inference stages, respectively, as well as 35\% and 64\% gains in time, while preserving the generation quality; details are provided in the Appenix, Section I.

---

> > > > ### Comment · Reviewer_5BMn · 2025-11-27
> > > >
> > > > The additional experiment is really appreciated and it does indeed make the paper stronger in my opinition. Upon reading Appendix I, I have two questions:
> > > > 1. It is stated that "[we] introduce corruption through another bimodal distribution". However, the corruption distribution in diffusion models is often a Gaussian. It is cetainly the case in [1]. I do not know how diffusion with a bimodal transition kernel works. Can the authors clarify?
> > > > 2. I am wondering why this experiment was done on CPU, rather than GPU? Appendix F suggests a GPU implementation is also available.
> > > >
> > > > [1] Ho, Jonathan, Ajay Jain, and Pieter Abbeel. "Denoising diffusion probabilistic models." NeurIPS 2020.

---

> > > > > ### Author Response · Authors · 2025-12-03
> > > > >
> > > > > We thank Reviewer 5BMn for their strong interest in the applications of our cLUT sampler.
> > > > >
> > > > > While our cLUT sampler is designed for arbitrary distributions, we aim to demonstrate its capabilities on models more challenging than a simple Gaussian. Although the original diffusion formulation indeed assumes Gaussian priors [1], extensions to non-Gaussian or data-dependent priors have been also explored. For example, Lee et al. [2] showed that reducing the mismatch between the data and noise distributions can improve model performance. Similarly, Nachmani et al. [3] demonstrated that sampling noise from a Gaussian mixture can enhance performance on certain tasks. Our experiments are consistent with these findings: when training on bimodal data, using Gaussian noise results in a substantially larger Wasserstein distance between generated samples and the training distribution (greater than 1), while using bimodal noise reduces this distance to below 0.07.
> > > > >
> > > > > Appendix F demonstrates not a full GPU implementation of our algorithm but rather its integration into the JAX library. CPU power measurement interfaces (e.g., RAPL) provide precise energy readings for accurate profiling and are therefore sufficient for algorithm-level comparisons.
> > > > >
> > > > >
> > > > > [1] Ho, Jonathan, Ajay Jain, and Pieter Abbeel. "Denoising diffusion probabilistic models", NeurIPS 2020.
> > > > > [2] Lee et al. "Priorgrad: Improving conditional denoising diffusion models with data-dependent adaptive prior", ICLR 2022.
> > > > > [3] Nachmani et al. "Non gaussian denoising diffusion models", arxiv preprint 2021.

---

### Official Review · Reviewer_PkgD · 2025-11-04

**Soundness:** 3
**Presentation:** 3
**Contribution:** 3
**Rating:** 6
**Confidence:** 2

**Summary:**

This paper proposes a new (to the best of my knowledge) method to generate random variables from arbitrary distributions, based on lookup tables. The approach is rather simple and well justified, and various benchmarks show a good improvement in time and energy savings with respect to state-of-the-art method.

**Strengths:**

The paper is clear, the approach is sound and simple (in a good way) while being theoretically justified.
Sampling is not a huge bottleneck in most modern applications of machine learning, but it is always good to make things more efficient.

**Weaknesses:**

I would have appreciated a more in-depth discussion of limitations. It does not seem clear to me that this will completely replace existing approaches, as it is less "out-of-the-box" than those.
Additionally, while you touch upon one Bayesian learning application, the analysis is quite light. It would be better to include a more in depth analysis for a more standard application (such as a more complicated distribution, eg sampled with MC methods for bayesian posteriors).

**Questions:**

See weaknesses.

---

> ### Author Response · Authors · 2025-11-20
>
> We thank the Reviewer PkgD04 for their careful assessment. The comments are taken into account to improve both our work and its presentation. In the following response, we address the questions and concerns raised.
>
> - **On "Limitations of cLUT deployment":**
> Thank you for noting that we should further elaborate on how cLUT compares to and can replace currently existing commonly used approaches. We did not emphasize this aspect in our paper because the core contribution is the introduction of a new algorithm rather than a new library.
>
>     We provide source code that allows cLUT to be used through a simple Python interface with only two lines of code. Given a probability vector`P`, cLUT can be constructed and used as follows:
>
>     ```python
>     clut_ctx = lut_c_wrapper.CLUTSampler(P) # preprocessing
>     samples_idx = clut_sample.sample_cLUT_fast(
>                 clut_ctx.cLUT.astype(np.uint32),
>                 clut_ctx.r, clut_ctx.c, N_SAMPLES
>             ) # sampling
>     ```
>     This code was used for algorithm benchmarking in the `python_experiment.py` file and for the TrueSkill evaluation in the`trueskill.py` file; both files are available in the provided repository. We also note that the other algorithms we benchmark cLUT against include their own preprocessing stage under the hood as well.
>
>     Alternatively, cLUT can be integrated into existing libraries. As an example, we demonstrate that incorporating cLUT into the JAX library can lead to persistent performance acceleration; details are provided in Section F of the Appendix.
>
>     Additionally, we provide data on the practical usage of cLUT for several common distributions (e.g., Normal, Gamma, and Exponential), including compression ratios and memory requirements; see Figure 6 in the Appendix.
>
>
> - **On "Bayesian learning applications"**:
> Thank you for raising the concern regarding the applications of cLUT. We would like to clarify why we included the TrueSkill example and why we believe it is sufficient.
>
>     While the primary evaluation of cLUT is performed independently of any specific application, we also provide the TrueSkill example to demonstrate the efficiency gains of a complete system when replacing a commonly used approach with our new cLUT. You suggested using a more standard application, such as sampling for Bayesian posteriors with a more complex distribution. However, the TrueSkill example is precisely such an application.
>
>     In our TrueSkill setting, two players share the same prior bimodal distribution (as an example of a non-conjugate prior) for their skills. After receiving the match outcome, posterior distributions for each player’s skill must be computed using an importance sampling scheme, as described in Section F of the Appendix. In this algorithm, cLUT replaces the standard sampling function, resulting in up to a 34% reduction in energy consumption.
>
>
> In conclusion, we would like to thank Reviewer PkgD04 for the valuable feedback. We hope that we have addressed the concern regarding out-of-the-box usage by demonstrating our Python interface, as well as the concern regarding a Bayesian learning application by explaining our TrueSkill example.

---

### Official Review · Reviewer_byEu · 2025-11-05

**Soundness:** 4
**Presentation:** 4
**Contribution:** 4
**Rating:** 8
**Confidence:** 3

**Summary:**

The paper introduces cLUT (compressed Lookup Tables), a method for exact, fast, and
energy-efficient random variate generation from arbitrary discrete distributions. The
key idea is to quantize the target distribution into integer frequencies, represent
those frequencies in binary, and compress the resulting lookup table by aligning entries
along bit-planes that correspond to powers of two. Sampling then reduces to drawing a
row index from a truncated geometric distribution and a column index uniformly from a
fixed-width table. This produces exact samples using only bit-level randomness and a
single memory access per draw.
The authors benchmark cLUT against classical exact samplers and demonstrate superior
performance on large discrete distributions in both speed and energy efficiency. They
further validate the method in an applied setting showing
that replacing NumPy’s RNG with cLUT yields equivalent posterior results at
substantially lower runtime and energy cost. Overall, the paper proposes an elegant
unification of algorithmic simplicity, theoretical exactness, and practical efficiency
for one of the oldest primitives in probabilistic computation.

**Strengths:**

- **Conceptual elegance and originality.**
  The compression of frequency tables along binary bit-planes is a simple yet powerful
  idea. It yields an exact, entropy-optimal sampling procedure that directly exploits
  the binary nature of digital hardware. The resulting geometric–uniform two-step
  sampling scheme is mathematically sound.

- **Clarity and quality of exposition.**
  The paper is very well written, with intuitive explanations and instructive figures
  (especially Figs. 1–2). The visual decomposition of frequencies into bit-planes makes
  the algorithm conceptually understandable.

- **Empirical thoroughness.**
  Benchmarks span both high-level (NumPy, PyTorch, JAX) and low-level (C)
  implementations, reporting not only runtime but also energy via RAPL counters. The
  TrueSkill case study convincingly demonstrates real-world impact, reducing both time
  and energy while preserving statistical equivalence of posterior estimates.

- **Practical significance.**
  Random number generation is a foundational operation across ML and simulation. A
  method that is *exact*, *faster*, and *more energy-efficient* without sacrificing
  correctness has immediate relevance for probabilistic modeling, simulation-based
  inference, and on-device ML. The bit-level efficiency arguments also connect nicely to
  ongoing discussions of energy-aware ML.

- **Implementation simplicity.**
  The algorithm relies only on two primitive RNG calls: a geometric distribution
  obtained from bit flips and a power-of-two uniform integer. Both are exact and
  hardware-friendly. This makes the method easy to embed in existing frameworks.

**Weaknesses:**

- **Limited discussion of integration and practical deployment.**
  While the method is described cleanly, the paper provides little guidance on how cLUT
  would be incorporated into standard frameworks such as `torch.distributions` or
  `jax.random`. In practice, the preprocessing cost and table storage would need to be
  managed per distribution instance or cached globally; a brief discussion of this would
  clarify the usability pathway.

- **Evaluation metric for TrueSkill comparison.**
  The comparison of posterior estimates between NumPy and cLUT uses t-tests on means.
  This only probes first-moment agreement; distributional similarity could be assessed
  more rigorously with two-sample metrics such as the **C2ST** score (which should
  approach 0.5 for indistinguishable posteriors).

- **Memory and break-even analysis.**
  Figure 5 shows clear scaling of preprocessing cost, but some contextualization would
  help. For very large supports (e.g., $n>10^8$), achieving the theoretical break-even
  might require enormous sample counts. Discussing typical discretization sizes for
  common continuous distributions (Normal, Gamma, etc.) would make the results more
  interpretable for practitioners.

- **Minor editorial points.**
  Line 235: stray capital ‘T’.
  Line 236: slight mix-up in text; should read “is replaced by two a in the third row,
  which corresponds to a frequency of 2.”

These are all minor and easily addressed; none affect the core soundness or clarity of
the work.

**Questions:**

1. **Construction for general distributions.**
   The cLUT method requires a discretized finite support to construct its compressed
   lookup tables. Could the authors clarify how this construction would proceed for
   *continuous or parameterized* distributions (e.g., those in `torch.distributions` or
   `jax.random`)?
   In particular:
   - How are truncation bounds and binning strategies chosen for unbounded or
     heavy-tailed distributions?
   - How is quantization precision $b$ selected in practice to balance fidelity and
     memory footprint?
   - Could this preprocessing be automated so that cLUT can serve as a backend for
     standard probabilistic libraries?

2. **Vectorization and hardware integration.**
   The reported benchmarks compare cLUT to scalar or lightly vectorized C baselines.
   Many ML frameworks already implement heavily vectorized sampling kernels.
   - Can the authors comment on how cLUT would integrate into such environments, would
     the same speedups hold once both systems are equally vectorized? For example:
      - Is the algorithm amenable to SIMD or GPU parallelization (drawing rows and columns
     in bulk and performing vectorized gathers)?
      - How does cLUT’s memory access pattern compare to typical GPU-friendly RNG kernels?

3. **Memory–compute trade-off and practical break-even.**
   Figure 5 illustrates a break-even analysis in terms of energy and wall-clock time,
   showing that cLUT’s higher preprocessing cost is amortized after sufficient draws.
   - Could the authors contextualize these numbers for typical workloads? For example,
     how many discretized bins are required to approximate a continuous distribution
     such as a Normal or Gamma to standard ML accuracy, and how many draws would that
     entail in realistic training or inference runs?
4. **Code availability and reproducibility.**
   Will the authors release an open-source implementation, preferably an anonymous
   repository for review, covering both CPU and GPU backends?
   Ideally, this should include scripts for rebuilding benchmark tables, integration
   examples with PyTorch or JAX, and automated evaluation of time and energy metrics.
   Such an implementation would be highly valuable for assessing portability and
   adoption potential.

---

> ### Author Response · Authors · 2025-11-20
>
> We thank the reviewer for their thoughtful and detailed assessment. We greatly appreciate the constructive suggestions and have incorporated multiple improvements into the revised version.
>
> - **On "Limited discussion of integration and practical deployment."**
> We agree that integration details are important. cLUT can be incorporated into existing frameworks either (i) by directly calling our C wrapper or (ii) via native framework implementations, since cLUT only requires access to Bernoulli random bits already provided by these libraries. To clarify this, we have added a paragraph in the Appendix, Section F, demonstrating how cLUT could be integrated into JAX’s `random.choice`, illustrating a minimal native implementation.
>
> - **On "Evaluation metric for TrueSkill comparison."**
> We would like to clarify our evaluation: as stated in Appendix G, we compared both means and variances, each aggregated over 50 independent runs. For each sampler, this yields two 50-dimensional vectors (means and variances), and we performed t-tests comparing these vectors across samplers, obtaining p-values > 0.2 in both cases, supporting the claim that there is no statistically significant difference in first and second moments.
> Regarding C2ST: while useful, it introduces dependencies on classifier architecture and hyperparameters. For this reason, we preferred tests with well-defined statistical properties and no additional modeling assumptions. We now mention this rationale explicitly in the revision.
>
> - **On "Memory and break-even analysis." and "Memory–compute trade-off and practical break-even."**
> Following the reviewer's suggestion, we added an  analysis (paragraph "Typical values" in the Evaluation Section 4, and Figure 6 in the Appendix, Section B) reporting typical cLUT parameter values, achieved compression ratios, memory usage, and discretization quality for common continuous distributions (Normal, Gamma, Exponential). We also note that our preprocessing code is preliminary and unoptimized. Optimized table construction would likely lower break-even points further.
>
> - **On "Minor editorial points."**
> We thank the reviewer for spotting these issues. All have been fixed in the revised draft.
>
> - **On "Construction for general distributions."**
> Appendix B provides an explanation of binning strategies. The lookup-table construction is fully automated in our implementation (see Appendix, Section D, for details). The choice of the quantization precision $b$ can also be automated, for instance, by selecting the smallest $b$ that achieves a user-specified bound on a distributional distance metric.
>
> - **On "Vectorization and hardware integration."**
> We have added a comparison of our cLUT approach integrated into the JAX library with the default sampling method from JAX. This experiment was run on a single A100 GPU, using JAXs internal mechanisms. We have demonstrated a persistent performance acceleration. Details are in the Appendix, Section F and Figure 7.
>
> - **On "Code availability and reproducibility."**
> We will release an anonymized, open-source repository for the review period. The repository includes automated evaluation of runtime and energy metrics, scripts for reproducing all benchmarks and figures, and an example integration of cLUT with JAX.
>
> We thank the reviewer for their extensive review, for pointing out weaknesses and for their helpful suggestions.
> We greatly appreciate the opportunity we were given to improve our paper.

---

> > ### Comment · Reviewer_byEu · 2025-11-21
> >
> > I thank the authors for answering my questions and commenting on the points made in my review!
> >
> > I have no further comments and maintain my evaluation that this is a good contribution to the ML research community.

---

### Author Response · Authors · 2025-11-21

We thank all reviewers for their careful reading, constructive feedback, and positive assessments. We have addressed each comment in detail in the respective responses.
Below we summarize the key improvements incorporated into the revised manuscript:

- Added a notation table for clarity (Appendix A).
- Included an analysis of typical parameter values for common distributions (Section 4, “Typical values” paragraph; Appendix B, Figure 6).
- Added detailed pseudocode for the distribute() function (Appendix D, Algorithm 3).
- Added an example integration into a popular ML library (JAX) (Appendix F).
- Added GPU evaluation for the JAX integration (Appendix F).
- Clarified the accuracy evaluation in the TrueSkill experiment (Appendix G).

We thank the reviewers again for their valuable feedback, which has significantly improved the clarity and completeness of the paper.

---

### Author Response · Authors · 2025-11-27

We thank all reviewers for their feedback and consideration of our improvements.

We thank Reviewer byEu for pointing out that cLUT (as well as ALDR and FLDR) do not increase substantially in runtime for distributions with $n\geq 10^5$, while the Alias method does.
This is due to increased sparsity (more and more outcomes having zero probability) in the distributions we evaluated on.
While cLUT, and ALDR and FLDR, handles sparsity well, common implementations of the Alias method do not.
Therefore, we adjusted the distributions we evaluate on to remove outcomes with zero probabilities in a first step before any evaluation, allowing for a more critical evaluation. Furthermore, we expanded the distributions set we test on to include distributions with $n\in[10^1,10^4]$, complimentary to the distributions with $n\in[10^4,10^8]$.
Our evaluations show that cLUT still substantially outperforms all other methods in terms of energy and time for all distribution sizes, including those with  $n\in[10^4,10^8]$.
We have moved the previous evaluation including sparse distributions to the Appendix, Section G.


Additionally, after receiving a question regarding an application of cLUT from reviewers 5BMn and PkgD04, we added an experiment evaluating the exemplary application of cLUT to a small-scale generative model, demonstrating cLUT's impact in a core ML problem highly relevant to the ICLR community.
Incorporating cLUT in a denoising diffusion model resulted in energy gains of 37% and 65% for training and inference stages, respectively, as well as 35\% and 64\% gains in time, while preserving the generation quality; details are provided in the Appenix, Section I.

---

### Author Response · Authors · 2025-12-03
**Summary of Revisions**

Short summary of main revisions:

- Added native JAX integration with GPU benchmarking showing consistent acceleration.
- Added a new application example: diffusion-model, demonstrating 35–64% time and 37–65% energy gains.
- Added complementary benchmarks across both sparse and dense regimes.
- Added full preprocessing pseudocode, a notation table, and clearer explanations of typical parameter values and memory usage.

We once again thank all reviewers for their careful reading, constructive feedback, and positive assessments.

---

### Meta-Review · Area_Chair_Mrwm · 2025-12-28

**Summary:**

The reviews are overall quite positive. Some minor concerns are raised, and the authors actively engaged to address the issues during the rebuttal. I suggest an accept.

**Reviewer Concerns:**

Several reviewers raised the concern of experiments, and it seems to me the authors have satisfactorily addressed this concern by updating the experiments.

There are also questions regarding the applications of cLUT. This is justified by new experiments to show the power of the method, which I find convincing.

Some reviewer also had concern with the missing time complexity analysis. This is still not covered by the rebuttal, although the reviewer eventually thinks this theoretical analysis is not necessary and hence considered solved. In my opinion, it is good to add a worst-case running-time analysis nonetheless.

Overall, I find the concerns of the reviewers are mostly addressed.

**Reviewer Scores:**

I think the reviewers will largely keep their scores unchanged.

---

### Decision · Program_Chairs · 2026-01-26

Accept (Poster)